# Artificial Intelligence in Predicting Microsatellite Instability and KRAS, BRAF Mutations from Whole-Slide Images in Colorectal Cancer: A Systematic Review

**DOI:** 10.3390/diagnostics14010099

**Published:** 2023-12-31

**Authors:** Theo Guitton, Pierre Allaume, Noémie Rabilloud, Nathalie Rioux-Leclercq, Sébastien Henno, Bruno Turlin, Marie-Dominique Galibert-Anne, Astrid Lièvre, Alexandra Lespagnol, Thierry Pécot, Solène-Florence Kammerer-Jacquet

**Affiliations:** 1Department of Pathology CHU de Rennes, Rennes 1 University, Pontchaillou Hospital, 2 Rue Henri Le Guilloux, CEDEX 09, 35033 Rennes, France; pierre.allaume@chu-rennes.fr (P.A.); nathalie.rioux-leclercq@chu-rennes.fr (N.R.-L.); soleneflorence.kammerer-jacquet@chu-rennes.fr (S.-F.K.-J.); 2Impact TEAM, Laboratoire Traitement du Signal et de l’Image (LTSI) INSERM, Rennes 1 University, Pontchaillou Hospital, CEDEX 09, 35033 Rennes, France; 3Department of Molecular Genetics and Medical Genomics CHU de Rennes, Rennes 1 University, Pontchaillou Hospital, 2 Rue Henri Le Guilloux, CEDEX 09, 35033 Rennes, France; marie-dominique.galibert.anne@chu-rennes.fr (M.-D.G.-A.); alexandra.lespagnol@chu-rennes.fr (A.L.); 4Department of Gastro-Entrology CHU de Rennes, Rennes 1 University, Pontchaillou Hospital, 2 Rue Henri Le Guilloux, CEDEX 09, 35033 Rennes, France; astrid.lievre@chu-rennes.fr; 5Facility for Artificial Intelligence and Image Analysis (FAIIA), Biosit UAR 3480 CNRS-US18 INSERM, Rennes University, 2 Avenue du Professeur Léon Bernard, 35042 Rennes, France

**Keywords:** digital pathology, artificial intelligence, colorectal cancer, deep learning, microsatellite instability, mismatch repair deficiency, *KRAS* and *BRAF* mutations

## Abstract

Mismatch repair deficiency (d-MMR)/microsatellite instability (MSI), *KRAS*, and *BRAF* mutational status are crucial for treating advanced colorectal cancer patients. Traditional methods like immunohistochemistry or polymerase chain reaction (PCR) can be challenged by artificial intelligence (AI) based on whole slide images (WSI) to predict tumor status. In this systematic review, we evaluated the role of AI in predicting MSI status, *KRAS*, and *BRAF* mutations in colorectal cancer. Studies published in PubMed up to June 2023 were included (*n* = 17), and we reported the risk of bias and the performance for each study. Some studies were impacted by the reduced number of slides included in the data set and the lack of external validation cohorts. Deep learning models for the d-MMR/MSI status showed a good performance in training cohorts (mean AUC = 0.89, [0.74–0.97]) but slightly less than expected in the validation cohort when available (mean AUC = 0.82, [0.63–0.98]). Contrary to the MSI status, the prediction of *KRAS* and *BRAF* mutations was less explored with a less robust methodology. The performance was lower, with a maximum of 0.77 in the training cohort, 0.58 in the validation cohort for *KRAS*, and 0.82 AUC in the training cohort for *BRAF*.

## 1. Introduction

### 1.1. Epidemiology

Adenocarcinoma is the most frequent histology for colorectal cancer. Europe has one of the highest incidence rates (28.8–32.1 per 100,000) after New Zealand and Australia [1,2]. The risk for colorectal cancer increases with age (median age 50 years old). Europe has a cumulative risk of 2–2.41% and 1.17–1.55% between 0 and 74 years for colon and rectum, respectively. Colorectal cancer is the second cancer in women and the third in men, making colorectal cancer a burden in Western countries [2].

### 1.2. Treatment

Depending on the stage, colorectal cancer has different treatments. When it turns out to be an invasive adenocarcinoma, surgical treatment is needed to remove a vast part of the intestine and its lymphatic nodes [3]. For advanced stages (II-III-IV), a non-surgical treatment (chemotherapy, radiotherapy, biologic therapy, immunotherapy, or any combination of these therapies) can be additionally proposed. The last guidelines for managing colorectal cancer emphasize the importance of biomarkers (MSI status, *KRAS*, and *BRAF* mutations) in advanced stages (stage II-III-IV). A delay of less than 28 days after surgery is recommended for the d-MMR/MSI status. In the case of an inoperable patient or metastatic situation, the status should be less than 14 days for the d-MMR/MSI status and *KRAS*, *BRAF* mutational status (Appendix A).

### 1.3. d-MMR/MSI Status

Detecting the d-MMR/MSI status enables patients to receive immunotherapy and the screening test for Lynch syndrome [4,5]. DNA mismatch repair (MMR) is the most important system for repairing DNA along with homologous recombination. Microsatellites are short DNA sequences of repeated nucleotides with a high probability of error. The DNA mismatch repair comprises the following genes: *MLH1*, *PMS2*, *MSH2*, and *MSH6*. The alteration of one of these genes results in a state named deficient MMR (d-MMR) and increases the probability of mutations in microsatellite regions [6,7]. A deficient MMR status has been reported in 5 to 20% of colorectal cancers [8]. Currently, to evaluate the deficient DNA mismatch repair, the techniques frequently used are immunohistochemistry of the four proteins (MLH1, PMS2, MSH2, and MSH6) in the MMR system or the polymerase chain reaction test (PCR) to detect microsatellite instability. The polymerase chain reaction test analyzes two mononucleotides (BAT-25 and BAT-26) and three dinucleotides (D5S346, D2S123, and D17S250) or five poly-A mononucleotide repeats (BAT-25, BAT-26, NR-21, NR-24, and NR-27) according to the centers [9]. An approach for the testing could be to screen the colorectal tumors using immunohistochemistry and, when a d-MMR status is detected, a molecular test such as the multiplex PCR coild be performed to confirm the MSI status [10].

### 1.4. KRAS and BRAF Mutational Status

There are multiple biomarkers used in therapeutic decisions (*KRAS* and *BRAF*). For instance, there is a correlation between a high MSI status and *BRAF* or *KRAS* mutation [11]. Although *BRAF* and *KRAS* are usually exclusive, rare cases of concomitant mutations were described [12]. In the case of *KRAS* mutations, patients are resistant to anti-*EGFR* therapy. *KRAS* mutations are well known and represent 42% of cases in Western countries [13]. The most common mutations of *KRAS* are located in codons 12 and 13 of exon 2, but there are also hot spot mutations in exons 3 and 4 [14]. The *BRAF* mutation in colorectal cancer (CRC) is associated with a poor prognosis [15]. The main hot spot mutation *BRAF V600E* is approximately present in 8% of CRC. Interestingly, contrary to melanoma, patients with *BRAF V600E* in their tumors seem to not respond to *BRAF* inhibitors [16]. However, the newest guidelines proposed the combination of anti-*BRAF* and anti-*EGFR* in the second line [10].

### 1.5. Artificial Intelligence Applied to Digital Pathology

Artificial intelligence has emerged as a transformative force in healthcare, especially in processing images from radiology to pathology. Indeed, in radiology, AI can be used to improve ultrasound image quality using denoising methods [17,18]. With the advent of digital pathology, it is now possible to process whole slide images (WSI) with computers [19]. Artificial intelligence (AI) is able to learn image features that are used to predict molecular status [20,21,22,23,24,25,26,27,28,29,30,31,32,33,34,35,36]. In pathology, the use of deep learning with WSI has created a new field of research called pathomics. More precisely, deep neural networks can be trained to predict a mutational status, known due to a molecular test, which provides the ground truth in machine learning from input WSI. In order to obtain good performance, AI needs a large amount of training data. Training is performed by comparing the output of a model iteratively over the training data with the ground truth and by updating the estimated weights in the model to minimize the error between the output and the ground truth. After training, the obtained AI model can be applied to new patients to predict their mutational status. A deep learning model approach can be separated into two distinct phases: in the developmental phase, the model is given a dataset and a ground truth (wanted outcomes) for training, testing, and validation purposes. Two validation methods may be used: either the developmental dataset is randomly split between a training, a validation, and a testing set, or cross-validation can be performed. This phase enables the fine-tuning of the algorithm but does not provide generalization. To that end, the model must be validated on an external validation dataset with different pre-analytic conditions, i.e., a dataset from another center or database. There are many key factors in making a performant deep learning model, one of the most critical lying in the constitution of the developmental dataset. This dataset must be large enough and representative of the targeted population. The “ground truth”, which serves as a reference for the algorithm, needs to be defined by precise, objective, if possible, multipara metric and consensual criteria. As such, “MSI status ground truth” may be defined via either immunohistochemistry, molecular biology, or both; the mutational status of KRAS and BRAF may be defined using next-generation sequencing. Proprietary software can finally be marketed if the results are robust enough and well generalized. Some algorithms are used for diagnostic purposes, such as detecting or grading tumors and quantifying biomarkers, such as the proliferation index (Ki67). Others are used to predict molecular alterations for therapeutic or prognosis.

### 1.6. Performance Evaluation

There are different metrics to evaluate the performance of algorithms. The area under the curve (AUC) of the receiver operating characteristic (ROC) is the most common. The ROC is obtained using increasing thresholds on predicted probabilities to define the sensitivity, the ratio between the true positives and the sum of true positives and false negatives, the false positive rate, and the ratio between false positives and the sum of false positives and true negatives. Performance can also be evaluated using the ratio between the correct predictions and the total number of predictions (accuracy (ACC)). No performance metric is recognized as superior, for each provides different information. Therefore, they are often used in conjunction. Since AUC is useful for evaluating the diagnostic ability of a binary classifier, it is of great interest for deep-learning-based histological models (diagnostic/screening purposes being the most common). However, it is to be noted that two models cannot be compared to one another through their respective performance metrics unless they have been validated on the same dataset.

### 1.7. Aim of the Review

The present study aims to provide a synthetic and comprehensive review of deep neural networksmodels for predicting the d-MMR/MSI status and BRAF/KRAS mutational status in colorectal cancer.

## 2. Materials and Methods

This review is a systematic review and follows the Preferred Reporting Items for Systematic Review and Meta-Analysis (PRISMA) statement (Appendix A) [37,38].

### 2.1. Inclusion and Exclusion Criteria

We searched for studies from PubMed using the following search terms:

(Artificial intelligence OR machine learning OR deep learning OR computer-assisted OR digital image analysis) AND (microsatellite instability OR MSI OR MMR OR mismatch repair OR molecular alterations OR *KRAS* OR *NRAS* OR *BRAF*) AND (whole slide image OR digital slides OR slide). All studies released up to June 2023 using artificial intelligence to predict the instability of microsatellites or other key mutations on WSI were included. If relevant, a manual selection could be performed. We excluded review articles, articles that were not published in English, and studies that were not related to colorectal cancer.

### 2.2. Data Extraction and Assessment of the Risk

For each article, we extracted information about the authors, years of publication, type of neural network, performance outcome such as area under the curve or accuracy, magnification, training set, validation set, and how the ground truth was established. To be more precise, one author extracted data from each study, and a second independent author validated the extracted data. The quality of all the articles was evaluated using the quality assessment of diagnostic accuracy studies (QUADAS-2) and shortened in a table (Appendix A) [39]. We applied the following criteria to stratify, as recommended by the QUADAS-2, all studies with a high risk. For domain 1 (patient selection), studies with only one data set were evaluated as high risk. Regarding domain 2 (index test), the absence of either a cross-validation or an external validation was considered a high risk. For domain 3 (reference standard), a high risk was considered if the ground truth was not specified or did not use a proper technique. To be classified as a “proper technique”, the PCR must have used a pentaplex PCR and/or immunochemistry must have used the set of the four proteins. We also evaluated every article using the PROBAST checklist (Appendix A) [40,41].

## 3. Results

### 3.1. Flowchart

The PubMed search allowed us to find 156 articles from 1999 to June 2023, Figure 1. A total of 136 articles were excluded from screening titles and abstracts, and 6 were removed after reading the full articles. A total of 11 were removed because they were literature reviews. We found 3 articles via manual reference checking and obtained 17 articles for systemic review (Figure 1). All studies used retrospective collected data sets on colorectal cancer. Among the studies, 14 only predicted the MSI status, 2 both the MSI and *BRAF/KRAS* status, and 1 of them focused on *KRAS*. All studies were summarized in Table 1.

### 3.2. Prediction of d-MMR/MSI Status

All the studies used retrospective data, and in the majority of the studies, the Cancer Genome Atlas (TCGA) was included as part of the study or the only data set (training or validation data set) except in four studies [23,25,32,35]. The ground truth for assessing the MSI/dMMR status was different from one cohort to another. In some studies, data were collected from different data sets, so the ground truth method, either IHC or PCR, could differ within one study [21]. Eight studies used PCR, five used IHC, three used both PCR and IHC, and one used NGS (next-generation sequencing). One study did not mention the technique that was used [29]. Within the same technique, there was a variation in the methodology: the number of proteins targeted by IHC and microsatellites targeted. For instance, the data set DATCH (Darmkrebs Chancen der Verhütung durch Screening) in Kather et al., Schrammen et al., and Echle et al. used a 3-plex PCR for its confirmation of the dMMR status. The NLCS data group (the Netherlands Cohort Study) in Echle et al. (2020;2022) and the DUSSEL data group (Dusseldorf, Germany) in Echle et al. (2022) used 2-plex IHC antibodies [19,22,27,34]. Concerning the validation, some studies (*n* = 5) split their data into training set/test set, most studies (*n* = 8) used cross validation and rare studies (*n* = 2) used both methods to compare their performances. Most studies (*n* = 11) used an external validation data set to predict the performance of the MSI/dMMR status. Only one study performed a blind validation on two data sets [36].

All the studies included were based on deep learning. Various architectures have been used for prediction: shuffle-net, ResNet-based, and Inception-V3 were the most frequently used. HE2RNA, MSInet, SLAM, MILwere, PPsnet, DeepSmile, MSIntuit, and Wise MSI were all used only once. Most studies (*n* = 15) have reported their performance metrics using the AUC (Figure 2); only a conference paper has published its results using accuracy [29].

Among the studies, Echle had included the largest number of slides (*n* = 6406 for one study and *n* = 8343 in the second) for model development (Shuffle Net or Resnet 18, depending on the study) [21,23]. They demonstrated encouraging results in both studies for the prediction of the MSI status with an AUC (area under the curve) of 0.96 for the external cohort. The cohort of Lee et al. used Inception-V3, which was trained on two data sets (TGCA and Saint Mary’s Hospital) and obtained one of the highest performances of all the studies for MSI status prediction on an internal cohort (0.97 on the SMH) [27]. However, the performance dropped when the model was only trained on the TCGA cohort and was validated on the SMH data set as an external validation (0.787).

### 3.3. Prediction of KRAS and BRAF Mutations

Some studies (n = 3) also explored the mutational status of *KRAS* and *BRAF*. The TCGA data set was used in two out of three studies, and the DACHS data set was used in the other study. To define the ground truth of biomarkers, NGS was used in two studies. One study did not specify which technique was used. All studies performed a cross-validation on the training data set, and only one used an external validation. Different deep learning models were used, such as Resnet, Inception V3, and SLAM. All performances were evaluated using the AUC. Schrammen et al. used the most data with DACHS (n = 2448) and obtained the best result to determine the *BRAF* mutation with an AUC of 0.82 [35]. All studies had almost similar results when determining the *KRAS* mutation with an AUC of 0.6 in Bilal et al. and Schrammen et al. and 0.58 in Jang et al. [31,34,35].

### 3.4. Assessment of the Risk of Bias and Applicability

With the QUADAS-2 tool, we were able to assess the reviewed studies (Figure 3). Most of the studies had at least one high-risk factor, and six studies did not have any high-risk factor. Regarding the studies about MSI status, nine studies had a high risk of bias in the patient selection. Out of 17 studies, 6 had a high risk of bias in the index test and 1 in reference standard studies. Of the three studies about *KRAS* and/or *BRAF* mutations, all had a high risk of bias in the patient selection, two out of three had a high risk in the domain index test, and only one had a high risk for the reference standard.

## 4. Discussion

### 4.1. Summary of the Review

The d-MMR/MSI and mutational status in colorectal cancers are crucial for the prognosis, therapy, and detection of Lynch syndrome [42]. Over the past few years, recent studies have emerged and shown how deep-learning-based tools could predict the MSI status and mutational status from WSI [43]. Park et al. published a systematic review of deep learning models for predicting microsatellite instability based on tumor histomorphology, comprising studies up to September 2021 [44]. However, publications on artificial intelligence in pathology are rapidly increasing; therefore, we aimed to provide an updated systematic review on the subject. Park et al.’s review included 13 studies, 8 of which are reported in the present study; the others were not available through our institution.

### 4.2. Present Review Limitations

Our systematic review does present some limitations. We extracted studies from only one database (PubMed) because our institution had no access to other databases such as Medline. Hence, we might have missed papers published in other journals. Our study follows the PRISMA guidelines, and therefore, we strived to provide an analysis of the risk of bias and assessment across our reviewed studies using the QUADAS-2 tool. [39]. This tool is well known and usually used in clinical research for diagnosis. However, the questions raised by QUADAS-2 (Appendix A) are not always applicable to AI studies. We also evaluated each study with PROBAST [41]. Unfortunately, this tool has been developed to evaluate the risk of bias and the applicability of predictive model studies and is better suited for clinical studies than AI studies. Another QUADAS tool under development, QUADAS-AI, might be more suited for all studies using deep learning, but it is still not published [45]. The use of specific checklists, such as the IJMEDI checklist, might also help improve the quality of AI-related studies, considering many of their specificities and raising the importance of ethical and ecological considerations [46].

### 4.3. Limits of Traditional Techniques, Interest in Non-Contributive and Discordant Cases and AI Tools

Since the “ground truth” is crucial in constructing a deep learning model, it is paramount to investigate how it is defined across studies. In evaluating a MSI status, both the PCR and IHC have common limitations regarding pre-analytic conditions, such as the cold ischemia time and fixation time, impairing the quality of the proteins and the DNA. Another common limitation is tumor heterogeneity. Inside the same tumor, cells can have different genetic and protein characteristics in all tumor parts. Concerning IHC, interpretation can sometimes be difficult due to the inflammation and/or the intensity of the nuclear labeling associated or not with background noise, making an inter-observer variability [47,48]. On the counterpart, the multiplex PCR also allows the detection of the MSI status with good sensitivity in the technical limit of a percentage of 10% of tumor cells [49]. IHC and molecular biology have a high concordance in detecting the d-MMR/MSI status but sometimes produce discordant results [50]. For such discordant cases, another assay is warranted with both IHC and molecular techniques, preferably on another tumor region, to counterbalance potential tumor heterogeneity. If the results remain discordant, a board of experts comprising molecular biologists, pathologists, and oncologists will be assembled to rule on the case. Artificial intelligence could help in these discordant cases as third-party testing to guide the medical’s decision to provide an oncogenetic consultation and closely monitor family members.

Next-generation sequencing is also impacted by tumor heterogeneity and DNA degradation over time. Moreover, the “mutational status” may be defined regarding one mutation only (for example, BRAF V600E) or all possible mutations of the targeted gene. This distinction may explain the difficulties in building an AI tool to predict mutational status such as KRAS status, for its mutations are spread across a wide range of exons, contrary to BRAF’s V600 hotspot. In the same spirit, there are many mutations of unknown significance. Including (or not) multiple mutations in the “ground truth” may complexity the model, but it might also increase its usefulness and should, therefore, be at least considered.

### 4.4. AI Approaches Bias and Applicability

The performance of AI is impacted by many adjustable variables such as the architecture, the type of sample, the number of slides in the cohort, the type of validation (internal vs. external), and the method to define the ground truth. One of the main problems is the size of the WSI (1.5 GB at high magnification), which prevents the whole slide from being analyzed at once. Most of the time, WSI are split into multiple tiles used as inputs of a convolutional neural network (CNN) to extract features (summary of all the information in the tile) that are used as inputs of a second neural network (NN) to predict, after multiple trainings, the mutational status. In addition, AI needs a huge number of high-quality slides to determine the tumor status with precision [47]. There are inherent biases, one of which is well known and called overfitting. Overfitting happens when a model cannot generalize a high performance obtained on the training data to other external data. That is why the presence or absence of an external validation is crucial. For example, in the study of Lee et al., the algorithm was only trained on the TCGA data set, and when they tried to validate their AI network, it showed a poor result (AUC of 0.787) [27]. AI has also shown the same limitations as usual techniques due to the amount of tumor tissue that can be analyzed. Most algorithms are developed on surgical specimens, allowing for better prediction due to the tumor representation. For all these reasons, there is a need to plan ahead in order to build clinically relevant deep learning models. Suitability between the goal and developmental dataset needs to be taken into account; for example, an algorithm developed on surgical specimens will not be suited to handle biopsies and present poor performances.

### 4.5. Radiomics Interest Alone and Combined with Pathomics

AI applied to radiologic images, called radiomics, was also evaluated to predict the MSI status in colorectal cancer with no exploration of *BRAF* and *KRAS* mutation status [51]. Contrary to pathomics, whole images in radiomics are smaller and consequently less informative but can be entirely used without preprocessing to train algorithms. Recent retrospective studies on the subject showed an AUC with a range from 0.78 to 0.96 AUC [51]. Some studies combined imaging with clinical and/or histological data (Ki-67, gender, age, tumor localization, differentiation degree of tumor, smoking history, hypertension, diabetes, and family history of cancer), allowing a better prediction of the MSI status [52]. The radiomics model had an AUC of 0.68, the clinical model had an AUC of 0.59, and the combining model had a better performance with an AUC of 0.75 but still inferior to pathomics results. Therefore, it might be of interest to build “multi-omics models” combining pathomics and radiomics

### 4.6. The Application of an Artificial Intelligence Tool

Interestingly, there is already an AI-based pre-screening tool for MSI detection in colorectal cancer that recently obtained CE-IVD authorization and is commercialized by Owkins and described in the Saillard et al. study [36]. The algorithm was trained on the TCGA data set and performed a high performance on an external validation on the PIAP (pathology AI platform) with an AUC of 0.97. They also performed a blind validation for the first AI tool based on d-MMR/MSI detection. To reach this result, they validated their model development on 600 anonymous slides on two different scanners (Philips and Roche) with different slide formats (isyntax and ndpi, respectively) from two different sites. The two sites with different scanners obtained very similar results with an AUC of 0.88 and 0.86. The MSIntuit reaches a sensitivity of 95% (CI: 95% [93–100%]) based on the IHC ground truth only and was able to rule out up to 40% of the slides from the PCR or IHC screening. However, prospective cohorts could be expected to integrate AI into screening guidelines.

### 4.7. Interpretability

Some histologic criteria were associated with a higher probability of MSI status, such as the presence of mucinous adenocarcinoma, signet ring cell carcinoma, medullary carcinoma, poorly differentiated adenocarcinoma, and Crohn’s-type inflammatory infiltration [53]. On the contrary, no morphologic patterns were described to be associated with *KRAS* and *BRAF* mutations except for mucinous adenocarcinomas, which were associated with a higher probability of *BRAF* mutation [54]. Deep learning studies outperformed morphology alone but are generally considered “black boxes”, which could represent a limitation in using these algorithms [55]. However, innovative approaches can identify the most predictive tiles. The interpretation of these tiles could help identify new morphological patterns associated with the MSI status [56]. The comprehension of the deep learning model is challenging and a prerequisite to its acceptance by clinicians. Pathologists have a crucial role in validating and interpreting the results of AI systems in pathology. The expertise of pathologists is essential to ensure the accuracy and reliability of AI-based diagnostics/prediction.

### 4.8. Artificial Intelligence in Routine and Ethics

With the emergence of new technologies, such as artificial intelligence, a novel approach to medicine is emerging. Predictive, personalized, preventive, and participatory medicine (P4) is a new kind of medicine, alimented by new approaches based on the emergence of algorithms [57]. The integration of AI into healthcare is increasing rapidly, and DNN-based models are seen as promising tools for image analysis. As stated earlier, understanding that the deep learning models are obscure, physicians must remain in control of their decisions [51]. In many studies, certain data sets, such as the Cancer Genome Atlas, are freely available to enhance the development of artificial intelligence algorithms (Table 1). One of the main limitations is the quality of the images and the associated lack of clinical information. Indeed, there is little information on data such as age, sex, and origin. We need to expand the datasets with significant geographical diversity to enable the algorithms to predict patients from developing countries [58]. Currently, the majority of algorithms are developed by companies and organizations based in developed countries. Furthermore, we need to develop robust algorithms with large data sets, external validation, and solid “gold standards” to avoid inter-observer variability [59]. For the d-MMR/MSI status, validation using immunohistochemistry and polymerase chain reaction is required.

## 5. Conclusions

In conclusion, artificial intelligence and particularly deep learning tools are holding great promise in healthcare. However, several challenges and considerations need to be addressed to ensure effectiveness and ethical integration into the clinical routine. Rigorous study design (various data sets and external validation), collaboration with healthcare professionals, and ethical awareness are needed. Deep neural networks are a promising approach to predicting the d-MMR/MSI status but are less performant for *KRAS* and *BRAF* mutations. To predict the d-MMR/MSI status, further prospective studies comparing AI results with traditional techniques are needed for use in routine activity.

## Figures and Tables

**Figure 1 diagnostics-14-00099-f001:**
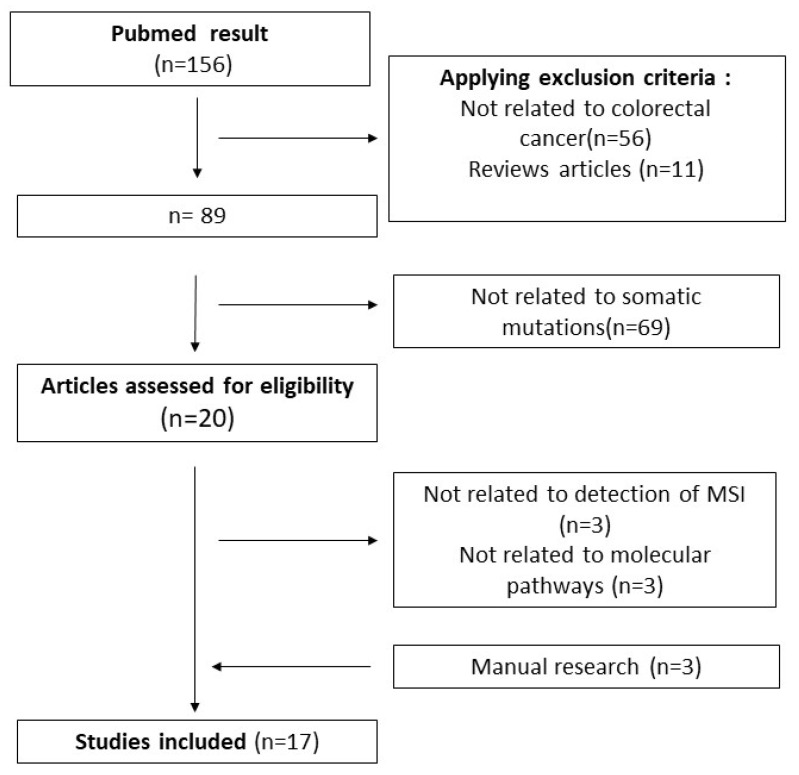
Flowchart.

**Figure 2 diagnostics-14-00099-f002:**
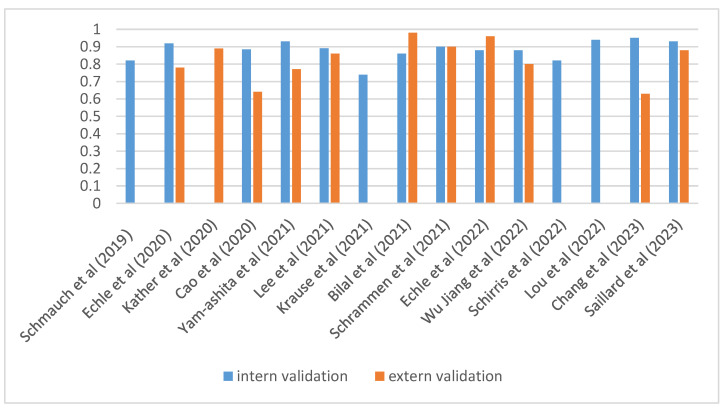
Overview of the AUC (area under the curve) for each MSI study between intern and extern validation [20,21,22,23,24,25,26,27,28,30,31,32,33,35,36].

**Figure 3 diagnostics-14-00099-f003:**
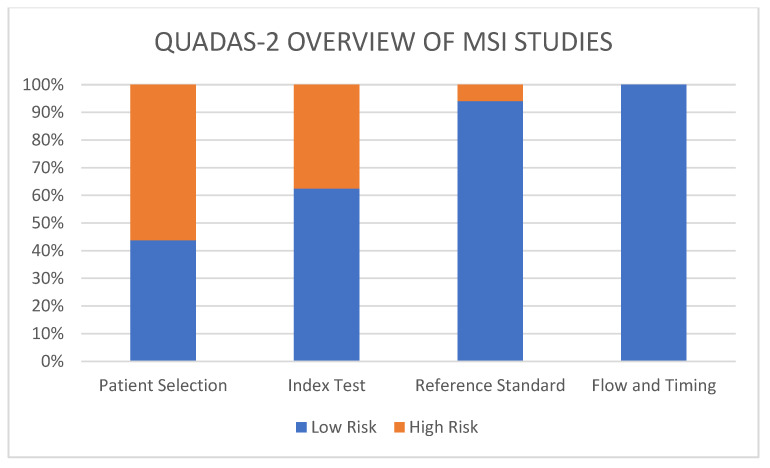
QUADAS 2 overview evaluating the risk of MSI studies.

**Table 1 diagnostics-14-00099-t001:** Comparison of studies.

Auteur	Year	Molecular Alteration	Data set	Neural Network	Magnification	Internal Validation	External Validation	Performance Metrics	Reference Molecular Status
Zhang et al. [29]	2018	MSI	TCGA	Inception-V3	x20 magnification	Random split	no	TCGA CRC Accuracy: 98.3%	not specified
			(CRC n = 585)						
Schmauch et al. [20]	2019								
		MSI	TCGA CRC FFPE	HE2RNA with ResNet50	x40 magnification	3-fold cross validation	no		PCR
			(n = 465 pts)					TCGA CRC FFPE: 0.82	
Echle et al. [23]	2020					random split	no	MSIDETECT CRC: 0.92 (0.90–0.93)	DACHS: PCR(1)
			MSIDETECT CRC	Shuffle net				MSIDETECT CRC: 0.92 (0.91–0.93)	TCGA: PCR
			(n = 6406 pts)			3-fold cross validation		YCR-BCIP-RESECT (n = 771 pts): 0.96	QUASAR and NLCS: IHC (3)
		MSI			not specified			(0.93–0.98)	YCR-BCIP: IHC
							yes	YCR-BCIP-BIOPSY (n = 1531 pts): 0.78	
								(0.75–0.81)	
			YCR-BCIP-BIOPSY			3-fold cross validation	no	YCR-BCIP-BIOPSY: 0.89 (0.88–0.91)	
			(n = 1531 pts)						
Kather et al. [28]	2020	MSI	TCGA CRC FFPE	ShuffleNet	x20 magnification	3-fold cross validation	yes	DACHS FFPE (n = 379 pts): 0.89 (0.88–0.92)	TCGA: PCR
			(n = 426 pts)						DACHS: PCR (1)
Cao et al. [26]	2020		TCGA-COAD Frozen			Random split	yes	TCGA-COAD: 0.8848 (0.8185–0.9512)	
			Total number including test cohort:					Asian-CRC FFPE (n = 785 WSIs): 0.6497	
			429 WSIs					(0.6061–0.6933)	
			TCGA-COAD Frozen (90%) +				no	Asian-CRC FFPE (n = 785 WSIs): 0.8504	TCGA-COAD: NGS (2)
			Asian-CRC FFPE (10%)					(0.7591–0.9323)	Asian-CRC: PCR
			TCGA-COAD Frozen (70%) +				no	Asian-CRC FFPE (n = 785 WSIs): 0.8627	
		MSI	Asian-CRC FFPE (30%)	ResNet-18	x20 magnification			(0.8208–0.9045)	
			TCGA-COAD Frozen (60%) +				no	Asian-CRC FFPE (n = 785 WSIs): 0.8967	
			Asian-CRC FFPE (40%)					(0.8596–0.9338)	
			TCGA-COAD Frozen (30%) +				no	Asian-CRC FFPE (n = 785 WSIs): 0.9028	
			Asian-CRC FFPE (60%)					(0.8534–0.9522)	
			TCGA-COAD Frozen (30%) +				no	Asian-CRC FFPE (n = 785 WSIs): 0.9264	
			Asian-CRC FFPE (70%)					(0.8806–0.9722)	
Jang, H.-J et al. [34].	2020		TCGA-COAD/TCGA-READ					TCGA:FFPE: 0.645(0.594–0.736)	
			n = 249				no	TCGA: Frozen: 0.778(0.675–0.937)	
		KRAS		Inception-v3 models	x20 magnification	10-fold cross validation			Sequencing
			SMH				yes	SMH: 0.58	
			n = 75						
Yamashita et al. [25].	2021					Random split	no	Stanford dataset (n = 15 pts): 0.931	Stanford dataset: IHC/PCR
								(0.771–1.000)	Four-fold TCGA: PCR
		MSI	Stanford dataset (n = 85 pts)	MSInet	x40 magnification				
						4- fold cross validation		Stanford dataset (n = 15 pts): 0.936	
							yes	TCGA (n = 479 pts): 0.779 (0.720–0.838)	
Lee et al. [27]	2021		TCGA FFPE					TCGA FFPE: 0.892 (0.855–0.929)	
			(n = 470,825 patches)				no	SMH FFPE: 0.972 (0.956–0.987)	
			SMH FFPE						
		MSI	(n = 274 WSIs)		x20 magnification				TCGA: PCR
			TCGA FFPE	Inception-V3		10-fold cross validation	yes	TCGA FFPE: 0.861 (0.819–0.903)	SMH: PCR/IHC
			(n = 470,825 patches)					SMH FFPE: 0.787 (0.743–0.830)	
			TCGA Frozen				no		
			(n = 562,837 patches)					TCGA Frozen: 0.942 (0.925–0.959)	
Krause et al. [24]	2021	MSI	TCGA FFPE (n = 256 pts)	ShuffleNet	x20 magnification	Random split	no	TCGA FFPE (n = 142 pts): 0.742 (0.681–0.854)	PCR
Bilal et al. [31]	2021	MSI						TCGA-CRC-DX: 0.86 (0.82–0.90)	PCR
			TCGA-CRC-DX				yes	PAIP: 0.98	
		BRAF	n = 499	Resnet 34	x20 magnification	4-fold cross validation		0.79 (0.78–0.80)	NGS
			PAIP				no		
		KRAS	n = 47					0.60 (0.56–0.64)	NGS
Schrammen et al. [35]	2021	MSI	YCR-BCIP					DACHS: 0.909 (0.888–0.929)	DACHS: PCR (1)
			n = 889				yes	YCR-BCIP: 0.900 (0.864–0.931)	PCR
		KRAS						DACHS: 0.609 (0.579–0.623)	not specified
			DACHS	SLAM	not specified	3-fold cross validation	no		
		BRAF	n = 2448					DACHS: 0.821 (0.786–0.852)	not specified
Echle et al. [21]	2022		DACHS					DACHS: 0.89 (0.87–0.92)	PCR(1)
			n = 2039						
			MUNICH					MUNICH: 0.88 (0.80–0.95)	IHC
			n = 287						
			TCGA					TCGA: 0.91 (0.87–0.95)	PCR
			n = 426						
			QUASSAR					QUASSAR: 0.93 (0.91–0.95)	IHC
		MSI	n = 1774	Resnet-18	not specified	8-fold cross validation	no		
			UMM					UMM: 0.92 (0.69–1.00)	PCR
			n = 35						
			MECC					MECC: 0.74 (0.69–0.80)	PCR
			n = 683						
			NLCS					NLCS: 0.92 (0.90–0.94)	IHC
			n = 2098						
			DUSSEL					DUSSEL: 0.85 (0.74–0.93	IHC
			n = 196						
			YORK SHIRE				yes	YORK SHIRE:0.96 (0.94–0.98)	IHC
			n = 805						
Wu Jiang et al. [22]	2022		TCGA					TCGA validation: 0.8888 (0.8531–0,9245)	
			n = 441						
			SYSUCC-surgical					SYSUCC-surgical: 0.8457 (0.8224–0.8690)	
		MSI	n = 355						IHC
			SYSUCC-biopsy	MIL	not specified	3-fold cross validation	yes	SYSUCC-biopsy: 0.7679 (0.7337–0.8021)	
			n = 341						
			PAIP					PAIP: 0.8806 (0.8574–0.9038)	
			n = 78						
Schirris et al. [30]	2022	MSI	TCGA-CR						
			n = 360	DeepSMile	not specified	Random Split	no	TCGA CR: 0.82 (0.77–0.86)	PCR
Lou et al. [32]	2022	MSI	Shandong Hospitals	PPsNET	x20 magnification	Random Split	no	Shandong Hospitals: 0.9429	IHC
			n = 144						
Chang et al. [33]	2023		TSMCC					TSMCC: 0.954 (0.94–0.96)	
		MSI	n = 1579	WiseMSI	not specified	10-fold cross validation			PCR
			TCGA					TCGA: 0.632 (0.703–0.733)	
			n = 609				yes		
Saillard et al. [36]	2023	MSI	TCGA					TCGA: 0.93 (0.90–0.96)	
			n = 859						
			PAIP	MSIntuit	not specified			PAIP: 0.97 (0.90–0.99)	PCR
			n = 47				yes		
			MPATH					MPATH-DP200: 0.88 (0.84–0.91)	IHC
			n = 600					MPATH-UFS: 0.86 (0.83–0.90)	

AUC, Area Under the Curve;TCGA, The Cancer Genome Atlas study; CRC, ColoRectal Cancer; WSI, Whole Slide Images; FFPE, Formalin-Fixed Paraffin-Embedded; DACHS, Darmkrebs: Chancen der Verhütung durch Screening (CRC prevention through screening study abbreviation in German); Stanford dataset, Stanford University Medical Center (USA); MSIDETECT: A consortium composed of TCGA, DACHS, the United Kingdom-based Quick and Simple and Reliable trial (QUASAR), and the Netherlands Cohort Study (NLCS); YCR-BCIP: Yorkshire Cancer Research Bowel Center Improvement Programme; SMH, Saint Mary’s Hospital (South Korea);TSMCC: TongShu MSI colorectalcancer;MPATH, medipath. 1–3-plex PCR (BAT25, BAT26, CAT25); 2-MSI sensor algorithm, 3- 2-plex IHC.

## Data Availability

Not applicable.

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
