# Peer review of "Artificial Intelligence in Predicting Microsatellite Instability and KRAS, BRAF Mutations from Whole-Slide Images in Colorectal Cancer: A Systematic Review"

_diagnostics, 2023, doi:10.3390/diagnostics14010099_

Round 1

Reviewer 1 Report

Comments and Suggestions for Authors

Applying artificial intelligence (AI) to predict cancer genetic mutation or biomarker status from whole slide images (WSI) of cancer tissue is playing an increasing important role in promoting AI pathology in precision oncology; in situations where no cancer tissue or cell is available for genetic testing; and in resource-poor developing countries where genetic testing for cancer is not available or limited 

Currently, pathologists are routinely testing dMMR and Braf by immunohistochemistry, and Kras mutation or other genetic mutation by NGS to guide oncologists to treat colorectal carcinoma. Pathologists are well aware that certain morphological features such as mucinous differentiation, medullary feature, poorly differentiated or Crohn’s like lymphocytic infiltrates of colon cancer are associated with higher rate of abnormal dMMR/MSI. Yet pathologists cannot predict accurately the dMMR deficiency status just from looking at histologic slides of colon cancer with human eyes

The authors correctly pointed out that the quality of data set, either online free digital images or clinical grade digital pathology slides; numbers of digital slides used; training models or training methods; and external validation are the key factors in determining the accuracy of the prediction or performance of the AI algorithm. Although some studies used free online TCGA digital images produced less robust prediction of the dMMR/MSI status of colon cancer as compared to clinical grade digital slides which performed better in predicting dMMR/MSI. By using advanced or sophisticated training models and solid external validation, free online TCGA digital images could be used to produce a commercial algorithm for dMMR/MSI prediction. AI digital pathology could play an important role in predicting dMMR/MSI of colon cancer.

Author Response

Dear reviewers,

The authors would like to thank you for the review of our article. We are grateful for your comments. An article was recently published (Validation of MSIntuit as an AI-based pre-screening tool for MSI detection from colorectal cancer histology slides, Charlie Saillard et al) and we added in our systematic review in the revised version.

Response to reviewer 1:

Thank you for your comments. Please find the modifications in red to the abstract, results, Figures and Tables due to the adding of the new article.

Respectfully,

Théo Guitton for all authors

Reviewer 2 Report

Comments and Suggestions for Authors

Regarding the manuscript titled "Artificial Intelligence for Predicting Microsatellite Instability and KRAS, BRAF Mutations from Whole-Slide Images in Colorectal Cancer: A Systematic Review":

In this systematic examination, the author scrutinized the utilization of artificial intelligence (AI) in predicting pivotal molecular characteristics (d-MMR/MSI, KRAS, and BRAF mutations) crucial for advanced colorectal cancer treatment. Employing 17 studies sourced from PubMed up to June 2023, the author evaluated both bias risk and performance, highlighting challenges such as a scarcity of slide numbers and the absence of external validation cohorts. Notably, deep learning models demonstrated commendable proficiency for d-MMR/MSI on training cohorts (mean AUC = 0.89), though exhibiting a slightly diminished performance on validation cohorts (mean AUC = 0.82). Conversely, the prediction of KRAS and BRAF, explored to a lesser extent, displayed reduced efficacy, with a maximum AUC of 0.77 for KRAS on training and 0.58 on validation, and 0.82 AUC for BRAF on training.

We wish to convey to the authors that this manuscript offers substantial value and successfully addresses an existing void. While the methodology is considered acceptable, possessing scientific merit and presenting quality content, we recommend revisions to enhance the study's overall quality. The following suggestions are provided for your consideration during the revision process:

1. Revise the structure of the introduction to establish a coherent flow. Commence with an introduction to challenges in the field of KRAS and BRAF mutations from whole-slide images in colorectal cancer. Subsequently, dedicate a paragraph to discuss the applications of artificial intelligence in healthcare, incorporating references to various studies in the field of image processing.

   - "Non-local adaptive hysteresis despeckling approach for medical ultrasound images."

   - "A Modified Adaptive Hysteresis Smoothing Approach for Image Denoising Based on Spatial Domain Redundancy."

   These references can provide a diverse perspective on artificial intelligence applications in medical image analysis, enriching your reference list.

2. Pay meticulous attention to the PRISMA protocol category to present the studies used more clearly. Specify the database for article extraction, elaborate on the stages of refining the articles, and draw inspiration from the article "Towards diagnostic aided systems in coronary artery disease detection."

3. Enhance the visual quality of the first image cover for a more comprehensive presentation.

4. Improve the legibility and organization of the first table.

5. Introduce comparative charts and statistics to address the study's lack, drawing inspiration from the provided article.

6. Strengthen the conclusion by expanding discussions for a more comprehensive summary.

In conclusion, while the study holds significant value, improvements are necessary for publication readiness. We encourage careful consideration of the provided suggestions for manuscript enhancement.

Author Response

Dear reviewer,

Response to reviewer 2:

  1. Revise the structure of the introduction to establish a coherent flow. Commence with an introduction to challenges in the field of KRAS and BRAF mutations from whole-slide images in colorectal cancer. Subsequently, dedicate a paragraph to discuss the applications of artificial intelligence in healthcare, incorporating references to various studies in the field of image processing. 

   - "Non-local adaptive hysteresis despeckling approach for medical ultrasound images."

   - "A Modified Adaptive Hysteresis Smoothing Approach for Image Denoising Based on Spatial Domain Redundancy."

Thank you for your insight. In the introduction, we began by the epidemiology and treatment and how the MSI and mutational status is crucial for the management of the patient. The challenging aspect of AI especially applied to prediction was more developed in paragraphs 4.3 and 4.4 of the discussion. We enlarge the paragraph 1.5 of the introduction to radiology and qote the references that you suggest.

  1. Pay meticulous attention to the PRISMA protocol category to present the studies used more clearly. Specify the database for article extraction, elaborate on the stages of refining the articles, and draw inspiration from the article "Towards diagnostic aided systems in coronary artery disease detection."

The database is Pubmed and specified in Materials and Methods and added on the Figure 1, flowchart. We also modified the results by describing more the Figure 1 in paragraph 3.1. We thank you for the reference that you gave us and consequently modified the flow chart.

  1. Enhance the visual quality of the first image cover for a more comprehensive presentation.

The Figure 1 was modified according to your recommendations.

  1. Improve the legibility and organization of the first table.

Table 1 was modified by sorting the studies by date and adding a recently published article (Saillard et al, ref 36).

  1. Introduce comparative charts and statistics to address the study's lack, drawing inspiration from the provided article.

Again, thanks to your reference, 2 Figures were added, one to highlight the different results (Figure 2) and one highlighting the different bias (Figure 3).

  1. Strengthen the conclusion by expanding discussions for a more comprehensive summary

The discussion was expanded according to recommendations. The present study was resituated in the known literature. More precisely, it was compared with Park’s et al. systematic review on MSI status prediction based on histomorphology from 2021

We discussed the management of discordant cases after both immunohistochemical and multiplex assays and their clinical implications. We discussed the use of the IJMEDI checklist for AI-based studies from Cabitza et al.

Paragraphs were reworked for better syntax. Orthographical mistakes were corrected.

Respectfully,

Théo Guitton for all authors

Round 2

Reviewer 2 Report

Comments and Suggestions for Authors

Dear [Author/Authors],

I extend my warmest congratulations to the authors of the manuscript titled "Artificial intelligence to predict Microsatellite Instability and KRAS/BRAF mutations from whole-slide images in colorectal cancer: a systematic thematic review."

The authors' meticulous attention to detail in addressing all revisions has significantly enhanced the overall quality of this study. The methodology employed is both acceptable and valuable, and the content presented is not only very good but also sufficiently comprehensive. It is evident that this handbook brings a novel contribution to the field, and I wholeheartedly recommend it.

I believe that the findings of this study will prove to be immensely useful for its intended audience. Your dedication to excellence is evident, and I anticipate that this manuscript will make a valuable impact in the relevant research community.

Once again, congratulations on your outstanding work.

Best regards,